# Rational Antibiotic Prescribing Is Underpinned by Dental Ethics Principles: Survey on Postgraduate and Undergraduate Dental Students’ Perceptions

**DOI:** 10.3390/antibiotics13050460

**Published:** 2024-05-17

**Authors:** Jelena Roganović, Milena Barać

**Affiliations:** Department of Pharmacology in Dentistry, Faculty of Dental Medicine, University of Belgrade, Dr Subotica 1, 11 000 Belgrade, Serbia; milena.barac@stomf.bg.ac.rs

**Keywords:** antimicrobial resistance, dental ethics, antibiotic prescribing, pharmacology

## Abstract

Background: Dentists bear the burden of responsibility for antimicrobial resistance since antibiotics are the drugs most prescribed by dentists. Often, “inappropriate” antibiotic use is considered as a “gray area” by dentists mainly due to ethical challenges associated with the clinical judgement depending on patients and/or prescribers. Aim: The study aimed to assess whether and in what way dental ethical principles underpin rational antibiotic use by investigating perceptions of postgraduate and undergraduate dental students without formal knowledge of dental ethics. Method: A cross-sectional anonymous survey comprised nine close-ended questions and was conducted among dental students (n = 125). The investigated practice of appropriate antibiotic prescribing in the survey relied on the respect of three basic principles of ethics: autonomy, non-maleficence, and beneficence. Results: Results show that dental students exhibit a lack of dental ethics knowledge that results in an inappropriate antibiotic-prescribing practice: prescribing an antibiotic when it is not necessary, without examination, or for indications that are not within the competence of the dentist. Multivariate regression analysis revealed that there was a significant difference between under- and postgraduates. Conclusions: Within the pharmacology course, a review of the clinical scenarios which cover both ethical and clinical complexities regarding the appropriate use of antibiotics should be introduced as an educational approach.

## 1. Introduction

The antimicrobial resistance represents a huge global problem, while the dental community bears the burden of responsibility since antibiotics are the class of drugs most prescribed by dentists. Recent evidence points at a regular, inappropriate use of antibiotics in dentistry [1,2]. The WHO defines appropriate antibiotic use as the cost-effective use of antibiotics aiming to maximize the clinical therapeutic effect while minimizing drug-related toxicity and associated with the development of antimicrobial resistance [3]. However, as far as inappropriate antibiotic use is concerned, dentists often describe it as a “gray area”: decision making under uncertainty, ethical challenges associated with the clinical judgement varying from patient to patient, depending on their vulnerability and risk population they are in, and clinical decisions that depend on the moral judgment of individual prescribers, varying in their antibiotic-prescribing approach [4,5]. So far, there is a huge number of articles proposing that a lack of adherence to the recommended prescribing protocol, based on choice of antibiotic appropriate for suspected pathogen, site of infection, dosing regimen, and duration of antibiotic therapy, represents the main obstacle for antibiotic stewardship in dentistry [6,7,8]. However, even when dentists follow guidelines, the overuse or misuse of antibiotics due to unresolved ethical issues can occur, and this is the problem that has not been addressed so far. Bearing this in mind, the aim of the present study was to assess whether and in what way dental ethical principles underpin rational antibiotic use by investigating the perceptions of postgraduate and undergraduate dental students without any formal knowledge of dental ethics.

## 2. Results

### 2.1. Autonomy

From 147 dental students invited to participate, 125 (85%) of them accepted to enter the study (Figure 1). 

Patients’ participation in decision making is considered ethical by 75.8% of postgraduates and 85.4% of undergraduates, while 96% of postgraduates and 100% of undergraduates consider giving patients all the required information, including those about side and adverse effects of drugs that may undermine compliance as completely ethical. However, as much as 67.7% of postgraduates and 77.7% of undergraduates consider giving a placebo, instead of an antibiotic, as an ethical practice in situations when there is no need for antibiotic use (Figure 2).

### 2.2. Non-Maleficence and Beneficence 

Prescribing an antibiotic when it is not necessary, e.g., in the case of non-complicated molar extraction in immunocompetent adults, is considered as ethical or ethical under some circumstances by 35.4% of postgraduates and even 58.7% of undergraduates, while 45.0% of postgraduates and 38.0% of undergraduates consider prescribing a broad-spectrum antibiotic, instead of the protocol-based recommendation of a narrow-spectrum antibiotic, as ethical, although almost all of them consider following the antibiotic-prescribing protocol as an ethical practice. Noteworthily, 17.7% of postgraduate and 12.6% of undergraduate dental students estimated antibiotic prescribing on patient’s or patient family’s demand as an ethical practice or ethical under some circumstances. Moreover, as much as 48.3% of postgraduates and 28.5% of undergraduates considered prescribing an antibiotic for a family member, without previous examination, as ethical, while around 38% of both postgraduates and undergraduates considered prescribing an antibiotic for a friend’s child with otitis as ethical or ethical under some circumstances (Figure 2).

### 2.3. Differences between Undergraduate and Postgraduate Students

Multilinear regression analysis of data revealed that there was a significant difference between under- and postgraduates only in responses to the question of whether it is ethical to prescribe antibiotics for a family member without examination, with a trend toward significance regarding responses to antibiotic use in the case of non-complicated molar extraction and to whether protocol should be followed. In these situations, where more postgraduates than undergraduates provided correct answers, explanation could be related to work experience rather than to knowledge of ethics (Table 1). Relationship between investigated dental students’ perceptions, represented as the covariance matrix between variables, is presented in Figure 3, while low levels of multicollinearity was obtained, represented by the Variance Inflation Factor (VIF) of <5 (Table 1). Analysis revealed that the investigated perceptions were relatively independent of each other, and thus, their association with the student status could be estimated with reasonable accuracy.

## 3. Discussion

The present results suggest that both postgraduate and undergraduate dental students show a lack of dental ethics knowledge that reflects in an inappropriate antibiotic-prescribing practice, such as prescribing an antibiotic when it is not necessary, prescribing a broad-spectrum antibiotic when it is not indicated, and prescribing an antibiotic without examination or for indications that are not within the competence of the dentist. Moreover, around 70% of our participants consider giving a placebo, instead of an antibiotic, as an ethical practice. The use of a placebo without the patient’s awareness could undermine trust, compromise the dentist–patient relationship, and result in medical harm to the patient. On the other side, the dentists should be aware that in situations where antibiotic prescribing is not indicated, the skillful use of reassurance for persistent patients is key to build respect and trust and improve health outcomes [9]. Although antibiotics are ineffective for toothache, or not always necessary for non-surgical interventions on teeth and periodontal tissues, antibiotic use in such cases is common [2]. Such inappropriate antibiotic use exposes patients to potential adverse effects and at the same time increases the risk of antimicrobial resistance development. One of the main reasons for the frequent prescription of antibiotics by dentists is to meet perceived expectations from patients and patients’ families, friends, or colleagues, and the present study showed that postgraduate and undergraduate dental students are not aware of the fact that antibiotic prescribing in order to satisfy patients’ or their family or colleague’s expectations represents a violation of several ethical principles. Patients are, in fact, satisfied if they perceive the dentist has shown interest and provided explanation for the pain or oral disease as well as the rationale for a specific dental treatment [10,11]. Ethical principles of beneficence and non-maleficence as well as professional integrity emphasize that the possibility of harm due to inappropriate antibiotic prescription take priority over patient’s or patient’s family or friend/colleague’s requests or financial gain. Another unethical practices, shown presently, are antibiotic prescribing as a (1) precautionary treatment, to prevent unlikely events of complications in immunocompetent patients, after non-surgical interventions, and which is different from antibiotic prophylaxis, given to patients at risk of developing bacterial endocarditis before the dental treatment, and (2) using broad-spectrum instead of protocol-recommended narrow-spectrum antibiotics. While broad-spectrum antibiotics may be a rational choice in situations where information is lacking about the source of an infection, broad-spectrum antibiotics are considered as strong drivers of antimicrobial resistance [12]. When it comes to a difference in perceptions between under- and postgraduates, it was found in responses to whether prescribing blanco prescriptions for a family member was ethical as well as in responses to the practice of prescribing antibiotic as a precautionary treatment. While more postgraduates than undergraduates provided correct answers regarding unnecessary antibiotic prescribing in a clinical setting, the explanation for this could lie in the work experience, since postgraduates lack familiarity with basic ethical principles as observed in other responses in the present survey. Bearing in mind that this is the first study showing that dentists, although keen to follow guidelines, may overuse and misuse antibiotics due to disrespect of basic ethical principles, i.e., patients’ autonomy, non-maleficence, and beneficence, we suggest that the respect of dental ethics principles is a core element of antibiotic stewardship, and resolving ethical issues related to prescribing antibiotics should be a part of the clinical pharmacology course in dentistry.

Dental ethics is an inseparable part of the dental practice [13] as the dentist has an ethical obligation to provide benefit to the patient, to avoid or minimize harm, and to respect the values and autonomy of the patient. Dental ethics has already established itself as an important and independent discipline structured on the problems specific for dentistry, and it differs from medical ethics in issues of professional goals as well as in the patient’s expectations: while professional goals in medicine emphasize health per se, dentistry needs to also achieve the patient’s psychological wellbeing including the fulfillment of patient’s desires [14,15]. Bearing this in mind, the expectations from medical doctors are to “care” about health, while expectations from dentists are also to “satisfy” the patient. These expectations thus may interfere with dentists’ attitudes and moral reasoning when prescribing antibiotics since infection treatment precedes other complex dental interventions aiming to reconstruct damaged oral functioning. Therefore, in medicine, patients are grateful if medical success is achieved, while in dentistry, a patient’s dissatisfaction could be related not just to oral health issues but rather to a disappointment due to failure in fulfilling his/her desires, or dissatisfaction with “purchased” dental works, such as dental crowns or dental implants [14]. Nevertheless, even if a patient insists, the dentist must not perform certain procedures, which could be harmful for the patient and compromise dentists’ professional reputation, such as an inappropriate antibiotic prescribing.

Thus, it is unethical to prescribe antibiotics when they are not indicated or simply as a precautionary treatment, with a broad- instead of the recommended narrow-spectrum antibiotic, as blanco prescriptions, or prescriptions for medical states other then dental, while regular informed consent process is obligatory. In the light of previously analyzed problem with the lack of dental ethics education at the Universities in Serbia [14], the present results show that dental students are not equipped to fulfill these ethical obligations and that their ethical skills must be improved in order to enable appropriate antibiotic prescribing. A dental ethics-oriented educational program should be included in under- and postgraduate dental studies, as separate or, at least as a part of the pharmacology course, aiming to improve awareness, knowledge, moral reasoning, and confidence of rational antibiotic prescribing. There is a suggestion that dental ethics education should start before clinical training, with engagement of different educational methods such as interactive mode and case studies [16]. Since there is a consensus that an alternative approach to didactic educational materials for antimicrobial stewardship is needed, we suggest case-based discussions involving ethical perspectives on antibiotic prescribing, such as real-world scenarios, peer-to-peer learning activities, and problem-solving exercises, to enhance learning in the pharmacology course. Also, the educators as moderators could provide a forum for examining and discussing complex clinical situations with students [17]. Furthermore, we have created and incorporated in the antibiotics studies a clinical decision support tool, mobile app. dentalantibiotic.com, aiming to prompt dental students to make informed decisions and adhere to best practices in antimicrobial stewardship. Education in dental ethics should also be continued after graduate studies, since continuing education in professional as well as in ethical behavior is necessary for maintaining competency in the upcoming era of digital and artificial intelligence-based tools [18], regardless of the dentist’s knowledge and skills.

## 4. Materials and Methods

A cross-sectional anonymous survey was conducted among postgraduate (PhD students and dental residents) and undergraduate students from the Faculty of Dental Medicine at the University of Belgrade (n = 125) (Figure 1). The survey comprised nine close-ended questions describing antibiotic-prescribing decisional problem associated with dental ethical consideration. The first draft of the questionnaire was prepared based on a review of published cases and vignettes on dental ethics, as well as interviews with practicing dentists [4,19]. Moreover, the trained bioethicists were invited to analyze the ethical implications of the questions and items. The feasibility of the questionnaire was then tested in a pilot study among a group of 50 dental students. As a measurement of the internal consistency of the questionnaire items, Cronbach’s alpha was estimated, and it was acceptable (0.71). As interviews revealed that participants exhibit a lack of dental ethics knowledge, a 3-point Likert-type scale was used [20], aiming to obtain straightforward responses from students in a “forced-choice” response format. For instance, participants were encouraged to respond carefully, and the proposed format reduced the response bias that could occur when students select the neutral option. We used the rank order scale due to the assumption that students could make relative judgments, even if they were not able to provide precise or accurate ratings. The survey was conducted for postgraduates (n = 62) and undergraduates (n = 63) on-site, and only after the undergraduates had successfully finished their pharmacology course in antimicrobials, by terms of successfully passing the pen-and-paper quiz regarding the use of antimicrobials in dentistry. Questionnaires were anonymous and voluntary. None of the participants had any formal education in medical/dental ethics. 

The study was conducted in accordance with the Declaration of Helsinki and was approved by the Ethics Committee of the School of Dental Medicine (approval number: 36/14; date: 10 March 2023). This cross-sectional survey used convenience sampling and was conducted between September and November 2023. The survey aimed to assess postgraduate and undergraduate students’ familiarity with ethical concerns of appropriate antibiotic prescribing, without any specific hypothesis testing. Thus, sample size estimation and power calculation were waived. The average time to complete the questionnaire was 7–10 min. The data were collected and managed using GraphPad Prism v.10. Descriptive statistics were presented as frequencies, and multivariate logistic regression analysis was used to investigate whether there was a relationship between responses to questions and status of the student (under- or postgraduate). A *p*-value of less than 0.05 was considered significant.

The investigated practice of appropriate antibiotic prescribing in the survey relied on the respect of three basic principles [14] (comprising three survey domains): (1) respecting *patient autonomy* in clinical decision making (Apply a placebo to a patient who requires an antibiotic, in situations where it is not indicated; in the case when the use of antibiotics is recommended and there is a choice of several antibiotics for that indication, consult the patient about the choice of drug; In the case of prescribing antibiotics, warn the patient about all possible side effects of the drug, even if they may threaten the patient’s motivation to use the drug); (2) not exposing patients to risks without a medical necessity—*non-maleficence* (Prescribe an antibiotic immediately after an uncomplicated and non-surgical extraction of the lower first molar in a healthy patient; Prescribe a broad-spectrum antibiotic in a situation where the protocol recommends the use of a narrow-spectrum antibiotic; To prescribe an antibiotic for a family member or colleague, without an examination); and (3) acting in the best interests of patients*—beneficence* (In your opinion, prescribe an antibiotic at the request of the patient or the patient’s family, in a situation where the use of antibiotics according to the protocol is not indicated; Prescribe an antibiotic to treat middle ear inflammation of your friend’s child; Prescribe antibiotics always and only according to the recommended treatment protocol). Nevertheless, inappropriate antibiotic-prescribing practice shown in the present study represents a violation of all mentioned ethical principles.

## 5. Conclusions

The present study revealed that both postgraduate and undergraduate dental students show lack of dental ethics knowledge that reflects in inappropriate antibiotic-prescribing practice, such as prescribing an antibiotic when it is not necessary, prescribing a broad-spectrum antibiotic when it is not indicated, and prescribing an antibiotic without examination or for indications that are not within the competence of the dentist. Thus, regardless of the existence of a separate course of dental ethics at dental schools, it would be preferable to introduce live classroom discussions in the pharmacology course regarding clinical scenarios that cover both ethical and clinical complexities regarding drug-prescribing practices and the appropriate use of antibiotics. It is essential for dentists to adhere to ethical principles along with following evidence-based guidelines of antibiotic prescribing, in order to minimize the risks associated with antibiotic resistance and adverse drugs reactions. 

## Figures and Tables

**Figure 1 antibiotics-13-00460-f001:**
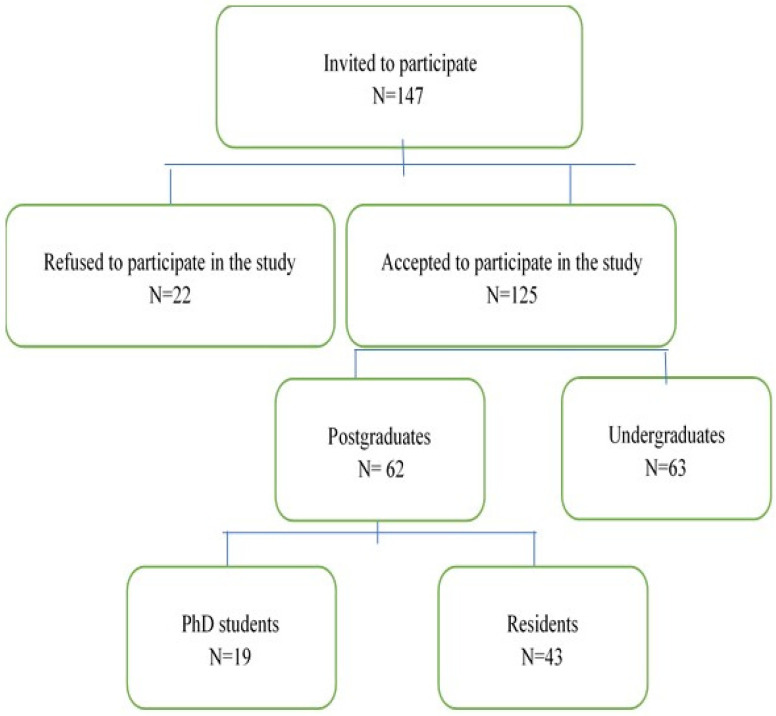
Study flowchart.

**Figure 2 antibiotics-13-00460-f002:**
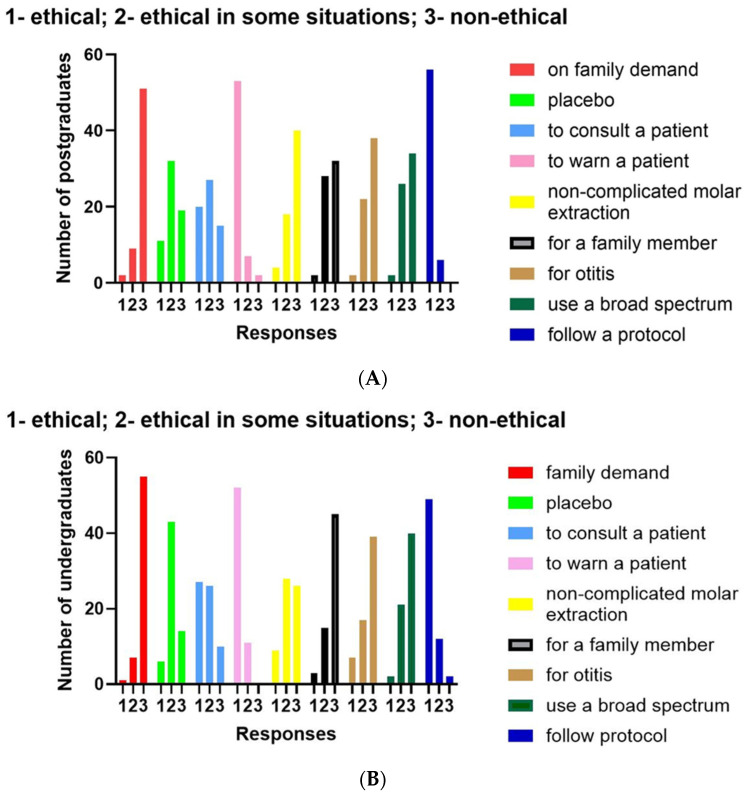
Distribution of responses to survey questions among postgraduate (**A**) and undergraduate (**B**) students.

**Figure 3 antibiotics-13-00460-f003:**
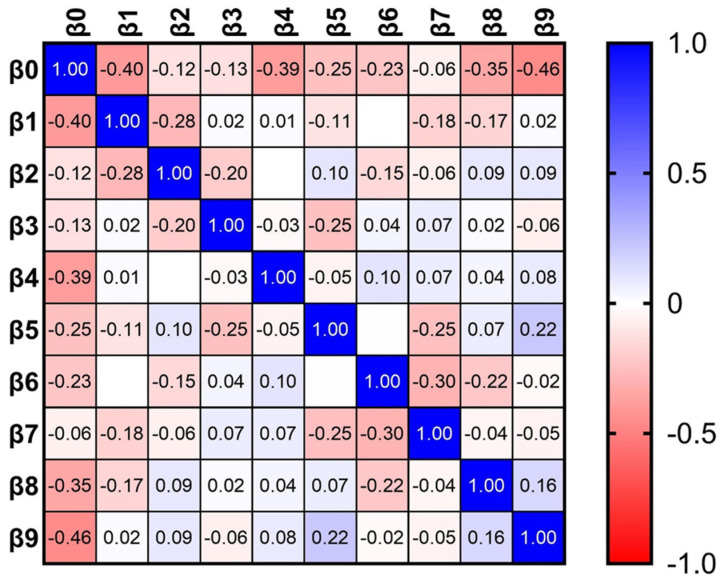
The magnitude of covariance between dental students’ perceptions. The higher absolute magnitude of covariance suggests a stronger relationship between variables (perceptions), while the sign indicates the direction of the relationship between variables.

**Table 1 antibiotics-13-00460-t001:** Relationship between dental students’ perceptions and status of the student (under- or postgraduate) obtained by multilinear regression.

Parameter Estimates	Variable	Coefficient (Estimate)	Standard Error	95% CI (Asymptotic)	|t|	*p* Value	VIF
β_0_	Intercept	2.05	0.45	1.16 to 2.94	4.57	<0.0001	
β_1_	Prescribe an antibiotic at the demand of the patient or the patient’s family, in a situation where the use of antibiotics is not indicated.	−0.14	0.11	−0.36 to 0.08	1.29	0.1991	1.24
β_2_	Administer a placebo to a patient who requires an antibiotic in situations where it is not indicated.	0.02	0.08	−0.13 to 0.17	0.24	0.8091	1.21
β_3_	When there is a choice of more than one antibiotic, one should consult the patient about the choice of drug.	0.06	0.06	−0.07 to 0.18	0.90	0.3697	1.11
β_4_	Warn the patient about all the possible side effects of the drug, even if this may threaten the patient’s motivation to use the drug.	−0.06	0.11	−0.27 to 0.15	0.56	0.5751	1.05
β_5_	Prescribe an antibiotic immediately after non-surgical extraction of the lower first molar in an immunocompetent patient.	0.14	0.07	−0.00 to 0.28	1.92	0.0573	1.23
β_6_	To prescribe an antibiotic for a family member or colleague without an examination.	−0.17	0.09	−0.34 to −0.00	1.99	0.0488	1.28
β_7_	To prescribe an antibiotic to treat your friend’s child’s otitis media.	0.09	0.08	−0.07 to 0.25	1.10	0.2724	1.33
β_8_	Prescribe a broad-spectrum antibiotic where the protocol recommends the use of a narrow-spectrum antibiotic.	−0.04	0.08	−0.20 to 0.13	0.47	0.6392	1.15
β_9_	Prescribe antibiotics always and only according to the recommended treatment protocol.	−0.20	0.11	−0.41 to 0.01	1.87	0.0639	1.09

Variance Inflation Factor (VIF) is used to detect multicollinearity among the independent variables. VIF values below 5 suggest low levels of multicollinearity.

## Data Availability

The data presented in this study are available on request from the corresponding author.

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
