# Peer review of "Rational Antibiotic Prescribing Is Underpinned by Dental Ethics Principles: Survey on Postgraduate and Undergraduate Dental Students’ Perceptions"

_antibiotics, 2024, doi:10.3390/antibiotics13050460_

Round 1
Reviewer 1 Report
Comments and Suggestions for Authors
The authors touches on the very important issue of antimicrobial resistance and irrational antibiotic prescription in dentistry and particularly aim to assess whether antibiotic use is supported by ethical principales by investigating perceptions of postgraduate and undergraduate dental students by the use of a short questionnaire. The results describe mainly the behavioural approach (which is of great importance) and to a lesser extent the scientific approach (what antibiotics are prescribed, for how long etc). The introduction is short and does not clarify whether similar surveys have been published . The analysis seems sound. The results obtained, though, do not present an add-on value on this issue, which is reflected in the conclusion, urging for more clinical pharmacology courses.
Comments on the Quality of English LanguageThe quality of English is good with a few flaws.
Author Response
RESPONSE TO REVIEWERS
We would like to thank the reviewers for constructive suggestions which have significantly improved our manuscript.
Reviewer 1
The authors touches on the very important issue of antimicrobial resistance and irrational antibiotic prescription in dentistry and particularly aim to assess whether antibiotic use is supported by ethical principales by investigating perceptions of postgraduate and undergraduate dental students by the use of a short questionnaire.
- The results describe mainly the behavioural approach (which is of great importance) and to a lesser extent the scientific approach (what antibiotics are prescribed, for how long etc). The introduction is short and does not clarify whether similar surveys have been published . The analysis seems sound.
Response: Accordingly, section Introduction has been modified.
“So far, there is a huge number of articles proposing that lack of adherence to recommended prescribing protocol, based on choice of antibiotic appropriate for suspected pathogen, site of infection, dosing regimen and duration of antibiotic therapy, represents the main obstacle to antibiotic stewardship in dentistry (https://doi.org/10.3390/ijerph20116025 doi: 10.1016/j.adaj.2019.08.020 /doi.org/10.1111/idj.12146 ). However, even when dentists follow guidelines, overuse or misuse of antibiotics due to unresolved ethical issues can occur, and this is the problem that has not been addressed so far”. And in Conclusions: “It is essential for dentists to adhere to ethical principles along with following evidence-based guidelines of antibiotic prescribing, in order to minimize the risks associated with antibiotic resistance and adverse drugs reactions”.
- The results obtained, though, do not present an add-on value on this issue, which is reflected in the conclusion, urging for more clinical pharmacology courses.
Response: Accordingly, we made clarifications in the section Discussion:
“Having in mind that this is the first study showing that dentists, although keen to follow guidelines, may overuse and misuse antibiotics due to disrespect of basic ethical principles: patients’ autonomy, non-maleficence and beneficence, we suggest that respect of dental ethics principles is a core element of antibiotic stewardship, and resolving ethical issues related to prescribing antibiotics should be the part of Clinical pharmacology course in dentistry”.
“Since there is a consensus that alternative approach to didactic educational materials for antimicrobial stewardship is needed, we suggest: case-based discussions involving ethical perspectives on antibiotic prescribing, such as real-world scenarios, peer-to-peer learning activities, and problem-solving exercises, to enhance learning in Pharmacology course. Also, the educators as moderators could provide a forum for examining and discussing complex clinical situations with students [17]. Furthermore, we incorporated clinical decision support tool: mobile app. dentalantibiotic.com aiming to prompt dental students to make informed decisions and adhere to best practices in antimicrobial stewardship”
Reviewer 2 Report
Comments and Suggestions for Authors
This is a qualitative study conducted at the University of Belgrade, Serbia to investigate dental students’ understanding of bioethics which impact on rational antibiotic prescribing.
The introduction is brief, but provides sufficient information on why the study was conducted: dentists describe rational antibiotic as “gray areas” that may be underpinned by dental ethical principles.
Because the methodology is placed near the end of the manuscript, it may be difficult for readers to grasp how the results are generated. The data seems to make more sense to me after reading the non-published materials (i.e. the survey) provided by the authors. This non-published survey seems to be a key to help understanding the methodology and result. Could this be included in the main manuscript?
The color code images are definitely very helpful for readers to read and interpret the figures. But for Figure 1, which questions belong to autonomy? Which questions belong to beneficence and non-maleficence? It is very difficult for readers to jump back and forth between figures and text to determine where the questions are from. Consider making this clear (e.g. which questions are to test autonomy/beneficence/non-maleficence?) in the figures and text.
Line 80-83 & Line 134-137: Why did it say there was a significant difference in response about molar extraction when P value was 0,0573; on the other hand, the question about “prescribe an antibiotic for a family member or col-league without an examination” has a P value of 0,0488 but was not considered significant?
Line 122: Not sure was that a typo. It said “that antibiotic prescribing should not be a determinant of antibiotic use”
Line 128: It seems like a run-on sentence here: “Another unethical practice is antibiotic prescribing as a precaution, to prevent unlikely events of complications even after non-surgical interventions, and then using broad spectrum antibiotics.” It seems to be describing two different points here: 1) prescribing as a precaution and 2) using broad spectrum antibiotics.
Line 160-161: It may be somewhat debatable to say prescribing antibiotics as a precaution is unethical. For instance, patients with high risk of endocarditis does require antibiotics as a precaution prior to dental procedures. There could be more invasive dental-maxillofacial surgeries that require antibiotic pre-op.
Line 166-167: This paragraph seems to overrate the impact of education to influence prescribing practice. As per the IDSA/SHEA guidance on implementation of antimicrobial stewardship program Section II (https://www.idsociety.org/practice-guideline/implementing-an-ASP/#RecommendationsAbridged), they “suggest against relying solely on didactic educational materials for stewardship.” Besides, these are students who may not have as much influence of prescribing as the practicing dentists.
The methodology has adequate info regarding setting, design, and research ethic. However, the authors should give clear definition of what undergraduate and postgraduate dental students mean. Do you mean they entered dental school after graduation from secondary school vs. graduation with a previous post-secondary degree? Or do you mean undergraduate BDS vs. postgraduate DMD, DDS, and PhD students? Giving some more information about this would help readers who are not familiar with the dental education structure.
The methodology could be even better if the authors could describe how the participants were recruited, excluded, and dropped out – this would help readers to determine the risk of bias in the participants’ responses.
The conclusion is adequate. It identifies key areas where dental students showed lack of dental ethics knowledge: prescribing an antibiotic when it is not necessary, prescribing a broad-spectrum antibiotic when it is not indicated as well as prescribing antibiotic without examination, and for indications that are not within the competence of the dentist.
Comments on the Quality of English LanguageRequire some editing.
Author Response
RESPONSE TO REVIEWERS
We would like to thank the reviewers for constructive suggestions which have significantly improved our manuscript.
Reviewer 2
This is a qualitative study conducted at the University of Belgrade, Serbia to investigate dental students’ understanding of bioethics which impact on rational antibiotic prescribing.
The introduction is brief, but provides sufficient information on why the study was conducted: dentists describe rational antibiotic as “gray areas” that may be underpinned by dental ethical principles.
- Because the methodology is placed near the end of the manuscript, it may be difficult for readers to grasp how the results are generated. The data seems to make more sense to me after reading the non-published materials (i.e. the survey) provided by the authors. This non-published survey seems to be a key to help understanding the methodology and result. Could this be included in the main manuscript?
Response: Accordingly, survey is included as supplemental material, but questions comprising it are listed in the Table and in Methods.
- The color code images are definitely very helpful for readers to read and interpret the figures. But for Figure 1, which questions belong to autonomy? Which questions belong to beneficence and non-maleficence? It is very difficult for readers to jump back and forth between figures and text to determine where the questions are from. Consider making this clear (e.g. which questions are to test autonomy/beneficence/non-maleficence?) in the figures and text.
Response: Accordingly, we modified Method section
“Investigated practice of appropriate antibiotic prescribing in the survey relied on the respect of three basic principles [14] (comprised three survey domains): 1) respecting patient autonomy in clinical decision-making (Apply a placebo to a patient who requires an antibiotic, in situations where it is not indicated; In the case when the use of antibiotics is recommended and there is a choice of several antibiotics for that indication, consult the patient about the choice of drug; In the case of prescribing antibiotics, warn the patient about all possible side effects of the drug, even if they may threaten the patient's motivation to use the drug.) 2) not involving patients to risks without medical necessity- non-maleficence (Prescribe an antibiotic immediately after an uncomplicated and non-surgical extraction of the lower first molar in a healthy patient; Prescribe a broad-spectrum antibiotic in a situation where the protocol recommends the use of a narrow-spectrum antibiotic; To prescribe an antibiotic for a family member or colleague, without an examination) 3) acting in the best interests of patients- beneficence (In your opinion, prescribe an antibiotic at the request of the patient or the patient's family, in a situation where the use of antibiotics according to the protocol is not indicated; Prescribe an antibiotic to treat middle ear inflammation of your friend's child; Prescribe antibiotics always and only according to the recommended treatment protocol). Nevertheless, inappropriate antibiotic prescribing practice shown in the present study actually represents, at the same time, a violation of all mentioned ethical principles”.
- Line 80-83 & Line 134-137: Why did it say there was a significant difference in response about molar extraction when P value was 0,0573; on the other hand, the question about “prescribe an antibiotic for a family member or col-league without an examination” has a P value of 0,0488 but was not considered significant?
Response: Corrections in the manuscript (results and Discussion) were made.
- Line 122: Not sure was that a typo. It said “that antibiotic prescribing should not be a determinant of antibiotic use”
Response: Accordingly, sentence is rephrased.
- Line 128: It seems like a run-on sentence here: “Another unethical practice is antibiotic prescribing as a precaution, to prevent unlikely events of complications even after non-surgical interventions, and then using broad spectrum antibiotics.” It seems to be describing two different points here: 1) prescribing as a precaution and 2) using broad spectrum antibiotics.
Response: Accordingly, corrections were made.
- Line 160-161: It may be somewhat debatable to say prescribing antibiotics as a precaution is unethical. For instance, patients with high risk of endocarditis does require antibiotics as a precaution prior to dental procedures. There could be more invasive dental-maxillofacial surgeries that require antibiotic pre-op.
Response. You are correct, however, when AB were indicated in risk patients, to prevent bacterial endocarditis, it is called Prophylaxis (AB were given before the risk treatment), and not Treatment. Precautionary treatment represents giving AB to healthy, after dental interventions, to remove the burden of responsibility if complications occur, which are often the result of negligence. Nevertheless, we rephrased the sentence in order to clarify that matter.
- Line 166-167: This paragraph seems to overrate the impact of education to influence prescribing practice. As per the IDSA/SHEA guidance on implementation of antimicrobial stewardship program Section II (https://www.idsociety.org/practice-guideline/implementing-an-ASP/#RecommendationsAbridged), they “suggest against relying solely on didactic educational materials for stewardship.” Besides, these are students who may not have as much influence of prescribing as the practicing dentists.
Response: Accordingly, we modified the Discussion section: “Since there is a consensus that alternative approach to didactic educational materials for antimicrobial stewardship is needed, we suggest: case-based discussions involving ethical perspectives on antibiotic prescribing, such as real-world scenarios, peer-to-peer learning activities, and problem-solving exercises, to enhance learning in Pharmacology course. Also, the educators as moderators could provide a forum for examining and discussing complex clinical situations with students [17]. Furthermore, we incorporated clinical decision support tool: mobile app. dentalantibiotic.com aiming to prompt dental students to make informed decisions and adhere to best practices in antimicrobial stewardship”
- The methodology has adequate info regarding setting, design, and research ethic. However, the authors should give clear definition of what undergraduate and postgraduate dental students mean. Do you mean they entered dental school after graduation from secondary school vs. graduation with a previous post-secondary degree? Or do you mean undergraduate BDS vs. postgraduate DMD, DDS, and PhD students? Giving some more information about this would help readers who are not familiar with the dental education structure.
Response: In Serbia, students enter the Faculty at the University, after graduating at Secondary school (19 years of age). Dental studies last for six years and afterwards, they can work as dentists or go on further studies: to be a specialist or to get a doctorate. Present study participants were undergraduates-third-year dental students attending Pharmacology course as well as postgraduates (residents or PhD students), as mentioned in the text.
- The methodology could be even better if the authors could describe how the participants were recruited, excluded, and dropped out – this would help readers to determine the risk of bias in the participants’ responses.
Response: Accordingly, flowchart is made and added as a figure.
The conclusion is adequate. It identifies key areas where dental students showed lack of dental ethics knowledge: prescribing an antibiotic when it is not necessary, prescribing a broad-spectrum antibiotic when it is not indicated as well as prescribing antibiotic without examination, and for indications that are not within the competence of the dentist.
Sincerely,
Jelena Roganović
Round 2
Reviewer 1 Report
Comments and Suggestions for Authors
The manuscrpit has been revised and the answers of the authors have addressed the points previously made.
Comments on the Quality of English LanguageI didn;t detect any flaws, although I am not a native speaker.
Reviewer 2 Report
Comments and Suggestions for Authors
The authors have addressed the points I raised. I do not have anything else to add.
Comments on the Quality of English LanguageNo major errors detected.